# Statistical survey of day-side magnetospheric current flow using Cluster observations: Bow shock

Evelyn Liebert[1], Christian Nabert[1], and Karl-Heinz Glassmeier[1]

[1]Institut für Geophysik und extraterrestrische Physik, Technische Universtität Braunschweig

*Correspondence to:* Evelyn Liebert (e.liebert@tu-bs.de)

**Abstract.** *We present the first comprehensive statistical survey of the day-side terrestrial bow shock current system based on a large number of Cluster spacecraft bow shock crossings. Calculating the 3-D current densities using Fluxgate Magnetometer data and the Curlometer technique enables the investigation of current locations, directions and magnitudes in dependence on arbitrary IMF orientation. In case of quasi-perpendicular shock geometries we find that the current properties are in good accordance to theory and existing simulation results. However, currents at quasi-parallel shock geometries next to the foreshock region underlie distinct variations regarding their directions.*

*Keywords*: Current systems (2708). Planetary bow shocks (2154). Instruments and techniques (2794).

## 1 Introduction

The terrestrial bow shock slows down the solar wind velocity to subsonic Mach numbers. This is accompanied by a gain in density, temperature and magnetic field strength. According to Ampère's law the bow shock carries electric currents which account for the jump in the magnetic field components tangential to the bow shock's surface. In contrast to the magnetopause, where the current directions are mainly determined by the geometry of the Earth's magnetic field, the direction of bow shock currents is solely determined by the orientation of the interplanetary magnetic field (IMF).

Depending on the local geometries of the shock surface and IMF orientation one distinguishes between quasi-perpendicular and quasi-parallel shocks where the IMF encounters the bow shock normal with angles above and below 45°, respectively. When encountering the compressed magnetic field at the shock some solar wind particles are reflected at the shock and re-accelerated in the solar wind's electric field while gyrating along the IMF direction before they enter the shock another time. At the quasi-parallel shock the reflected particles form the foreshock region where upstream waves are generated and alter the magnetic field configuration.

Until today, detailed bow shock current analysis based on in situ measurements has barely been done. Tang et al. (2012) presented a first statistical survey of bow shock currents using Cluster data from 25 crossing events when the IMF was dominated by its $B_z$ component. They selected quasi-perpendicular shocks near the bow shock nose and calculated the current density from the magnetic gradient and the shock thickness. Recently, Hamrin et al. (2017) investigated the currents of 154 quasi-perpendicular bow shock crossings near the shock nose using data from the Magnetospheric Multiscale (MMS) mission. In this paper we extend the statistical survey of bow shock currents at larger distances to the bow shock nose to overall 369

events covering both quasi-perpendicular and quasi-parallel situations during arbitrary IMF configurations for the first time. Making use of the simultaneously collected magnetic field data supplied by the multi-spacecraft mission Cluster (Escoubet et al., 2001) and applying the Curlometer technique (Dunlop et al., 1988) allows a direct 3-D investigation of the local current density vector.

## 2   Data selection and preparation

For our investigation we use Cluster magnetic field data from the Fluxgate Magnetometer (FGM) (Balogh et al., 2001) at spin resolution (0.25 Hz). Additionally, data from the Cluster Ion Spectrometry (CIS) instrument (Rème et al., 1997) are used to support the identification of bow shock crossings. The data are retrieved from the Cluster Active Archive (Laakso et al., 2010). Cluster consists of four individual spacecraft orbiting along a polar orbit with relative separations of a few kilometers up to over 10000 km. The thickness of the Earth's bow shock is about 100 to 1000 km. In order to match these spatial dimensions we use data obtained during periods when the average inter-spacecraft distance was small at the position of the bow shock. This criterion is fulfilled in the time range from February to May 2002 and from December 2003 to May 2004 when the average inter-spacecraft distance is about 300 km or less. At these times FGM and CIS data are available at the bow shock for 162 inbound and outbound orbit segments.

The Curlometer technique estimates the local current density across the Cluster tetrahedron volume by appoximating Ampère's law $\nabla \times \boldsymbol{B} = \mu_0 \boldsymbol{J}$. A thorough introduction to the Curlometer technique and an error analysis focussing on our application can be found in our previous publication (Liebert et al., 2017) where we conducted a similar study on magnetopause currents. The reliability of Curlometer results depends on the Cluster tetrahedron geometry which is constantly changing along its trajectory. One possibility to characterize the shape of the tetrahedron is given by the quality factor $Q_G$, which is defined by

$$Q_G = \frac{\text{True Volume}}{\text{Ideal Volume}} + \frac{\text{True Surface}}{\text{Ideal Surface}} + 1 \tag{1}$$

(e.g. Glassmeier et al., 2001), where the ideal volume and surface represent the volume and surface of a perfect regular tetrahedron with a side length equal to the average side length of the true tetrahedron. The quality factors equals one when the true tetrahedron is deformed into a linear geometry and three in case the true tetrahedron equals the ideal one. In this study we limit our investigation to bow shock crossings, where the quality factor takes values of 2.5 or more, leaving us 111 orbit segments. The usage of this quality factor for our investigation is discussed in detail by Liebert et al. (2017). $Q_G \geq 2.5$ allows us to expect accuracies of at least 2° to 10° in direction and 3 % to 15 % for the relative error in magnitude.

Dunlop et al. (2001) pointed out that high frequency fluctuations are likely to cause uncertainties within the Curlometer results and suggest an appropriate averaging in time before applying the Curlometer to magnetic field data. Comparing the effect of averaging windows of different sizes on the events investigated in our study prove the current directions to be quite insensitive to a window size between 20 and 40 s. Significant alteration of the directions sets in for windows below 10 s and above about 60 s as the influence of spatial scales smaller and larger than the bow shock scales risees. Current magnitudes

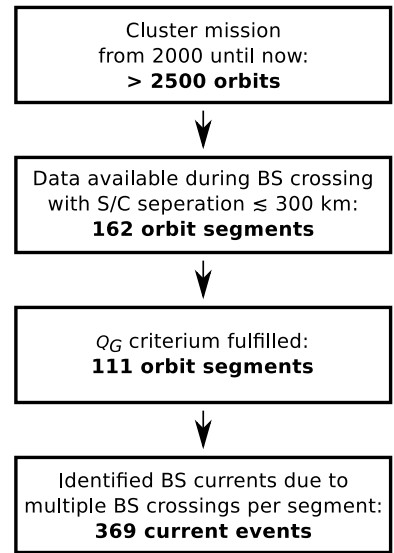

**Figure 1.** This sketch of the data selection process shows the shrinking amount of suitable data sets. From over 2500 Cluster orbits only a few orbit segments match the spatial requirements and the quality criterion for our bow shock current investigation.

are more sensitive to the size of the averaging window. The damping of high frequency fluctuation and the associated spatial averaging directly lead to smaller current peak magnitudes. This effect is less intense when an average magnitude per event is calculated instead. For the statistical study presented here, we chose a 30 s averaging window, which proved to be sufficient to damp highly fluctuating current signatures with spatial dimensions far below the tetrahedron size without altering current
structures having dimensions of about 50 km and more along the spacecraft trajectory to a significant extent.

After the application of the Curlometer tool, we look for bow shock current events by visual inspection of the Curlometer results. Currents are identified as bow shock currents when the following criteria are fulfilled: 1) a clear current peak is visible, and 2) the current event coincides with particle data signatures that are consistent with a bow shock crossing (see example event in Fig. 2). At each transition event the edges of the corresponding current feature are identified. We calculate the average
current directions and magnitude for every event. There are cases where the current or the particle data or both show very fluctuating behavior making it difficult to identify current structures at a bow shock crossing. In cases, where the identification becomes presumably unreliable, the events are omitted and not included in our study.

Caused by relative movement of the bow shock with respect to Cluster often multiple crossings in a row are recorded along an orbit segment. This enlarges our database for the statistical survey to 369 current events. Fig. 3 gives an overview of the
locations of all events in GSE-coordiantes.

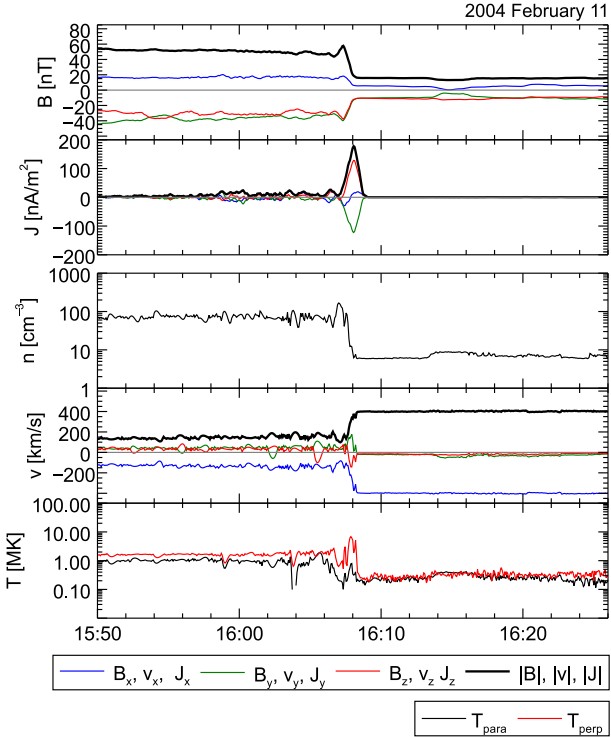

**Figure 2.** Example of a bow shock crossing during an outbound orbit on 11 February 2004. The upper and middle panels show Cluster 1 magnetic field data and the Curlometer results in GSE coordinates after application of a 30 s averaging window. The lower panel shows Cluster 1 CIS hot ion density $n$, velocity $v$, and temperature $T$. At 16:08 UT the transition from magnetosheath plasma (left-hand side) into undisturbed solar wind plasma (right-hand side) is visible within the particle data as well as within the magnetic field data. During the bow shock corrsing a clear current density peak is seen.

## 3 Reference bow shock

The position and size of the bow shock vary depending on the solar wind conditions to a large extend. For representation of the bow shock crossing locations we introduce a parabolic model reference bow shock as a common frame of reference. Following Nabert et al. (2013) we use the parametrization

$$x = \Delta_{\mathrm{BS}} - \sum_{t=y,z} c_{\mathrm{BS},t}\, t^2 \quad . \tag{2}$$

$\Delta_{\mathrm{BS}}$ depicts the sub-solar bow shock stand-off distance with respect to the centre of the Earth (see Fig. 4). The geometric parameters $c_{\mathrm{BS},t}$ represent the bow shock curvature in $t = y$ and $t = z$ direction. Nabert et al. (2013) deduce values of

$$c_{\mathrm{BS},y} = 0.4 \frac{1}{\Delta_{\mathrm{BS}}} \quad , \quad c_{\mathrm{BS},z} = 0.5 \frac{1}{\Delta_{\mathrm{BS}}} \tag{3}$$

from an analytical zeroth-order approach to solve the MHD-equations in the magnetosheath.

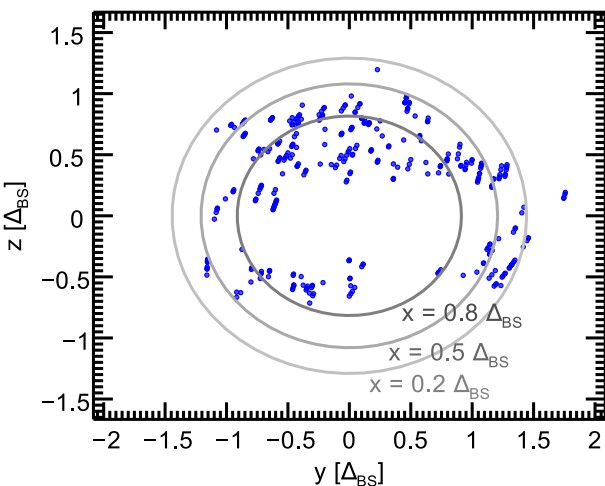

**Figure 3.** Location of the investigated bow shock crossings projected onto the reference bow shock presented in a GSE coordinate system. The grey ellipsoids represent the bow shock position at $x = 0.2, 0.5$ and $0.8\Delta_{BS}$ with $\Delta_{BS}$ depicting the bow shock stand-off distance.

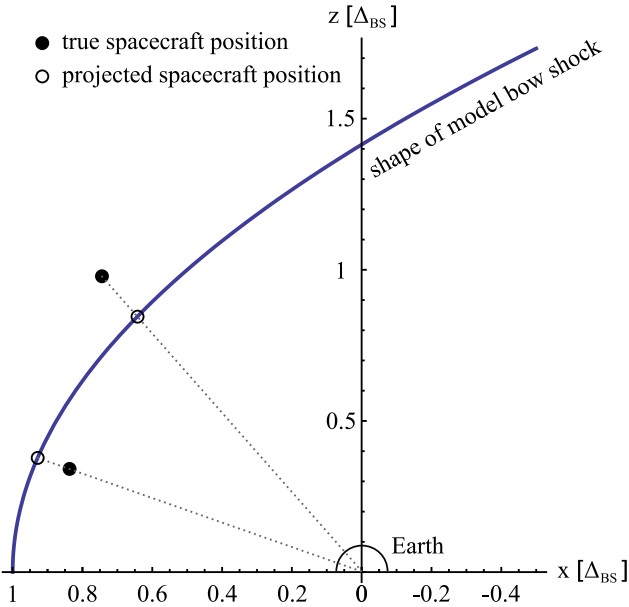

**Figure 4.** Sketch of the model bow shock used as a reference bow shock for data presentation. The dotted lines show example projections of two different currents from their true location to the location at the reference bow shock.

For each identified current the mean value of the tetrahedron barycentre's position vector is calculated. By radial projection along the Earth-spacecraft-line the intersection of this vector with the reference bow shock is calculated (cf. Fig. 4). Fig. 3 gives an overview of the location of all events after projection onto the reference bow shock within GSE-coordiantes.

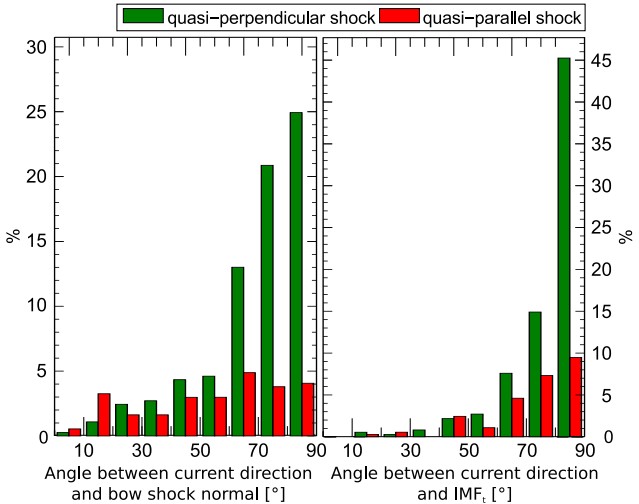

**Figure 5.** Frequency distribution of the angle difference between the current directions and the local bow shock normal (left) and the IMF tangential component (right). The angles are binned with $10°$-intervals.

Current directions at the bow shock are directly controlled by the IMF orientation via Amperè's law. As we do not confine our study to mainly north-south orientated IMF, the presentation of the resulting current directions in a GSE-system like in Fig. 3 would lead to a quite chaotic looking distribution and would make it almost impossible to extract useful information from it as the required information about the IMF orientation would not be included in such a picture. In order to account for the varying IMF orientations we conduct a second step of transformation by rotating the coordinate system around the GSE $x$ axis in such a way that the $IMF_{yz,GSE}$ component is orientated in positive $z$ direction within the new IMF-aligned coordinate system. The IMF is calculated by averaging the magnetic field data obtained during 5 minutes ahead of each bow shock current event.

## 4  Results

### 4.1  Directions of bow shock currents

Depending on the angle $\Phi$ between local shock normal and IMF, current events are categorized into quasi-perpendicular ($\Phi > 45°$) and quasi-parallel shocks ($\Phi < 45°$). The majority of 274 of the investigated currents represent quasi-perpendicular geometries. It is likely that this imbalance is caused by our event selection procedure. Within the foreshock region of a quasi-parallel shock oscillations are triggered and develop while the plasma is convected towards the shock surface. This causes fluctuations within the particle and the magnetic field data, leading to less clear plasma transition and current signatures. As described in section 2, events are omitted, when a reliable bow shock current identification is not possible.

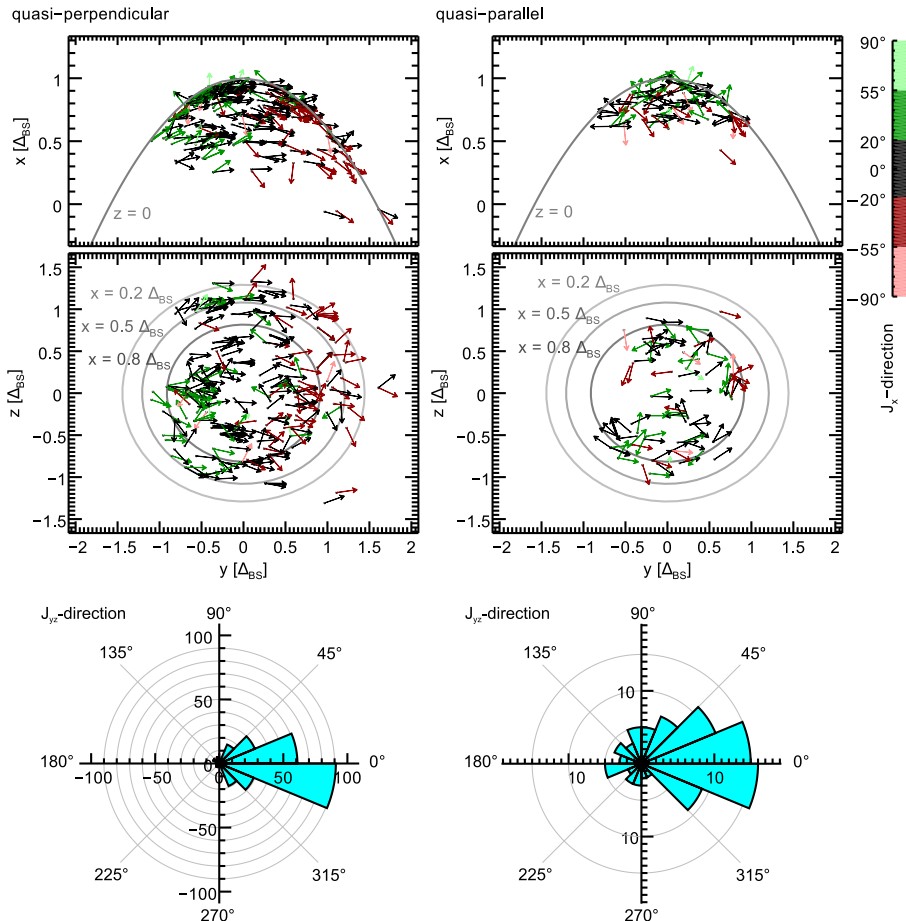

**Figure 6.** Bow shock current directions in *x-y* plane (top panel) and *y-z* plane (middle panel). $J_x$ direction is represented by the colour code. The *z* axis of the coordinate system is aligned to the $IMF_{yz}$ component. The grey paraboloid and ellipsoids represent the bow shock position at $z = 0$ and $x = 0.2, 0.5$ and $0.8\Delta_{BS}$, respectively. Polar histograms (bottom panel) show the occurrence rates of current angles within in the *y-z* plane with respect to the positive *y* axis.

Fig. 5 shows the orientation of the current flow with respect to the model bow shock normal and the IMF. Most of the observed bow shock currents during quasi-perpendicular geometries lie nearly perpendicular to both the shock normal, which means that the currents flow parallel to the bow shock surface, and the IMF tangential component as it is expected from theory. The deviation from perpendicularity with respect to $IMF_t$ is much larger in case of quasi-parallel shock situations (95 events) which reflects more turbulent and fluctuating conditions of the plasma flow adjacent to the foreshock region in contrast to the quasi-perpendicular shock. The broad distribution of the angle between current direction and reference bow shock normal indicates that the current flow direction at the quasi-parallel bow shock deviates extremely from the shape of a simplified bow shock surface.

**Table 1.** Occurrence rate of outward and inward pointing directions of the current normal component at the quasi-perpendicular bow shock depending on the $y$ coordinate within the IMF-aligned coordinate system.

| event location | $y < -0.5\,\Delta_{\mathrm{BS}}$ | $-0.5\,\Delta_{\mathrm{BS}} < y < 0$ | $0 < y < 0.5\,\Delta_{\mathrm{BS}}$ | $0.5\,\Delta_{\mathrm{BS}} < y$ |
|---|---|---|---|---|
| $J_n$ pointing outwards | 38 (48%) | 36 (61%) | 37 (57%) | 47 (67%) |
| $J_n$ pointing inwards | 46 (55%) | 29 (39%) | 28 (43%) | 27 (36%) |

The global current direction distribution of the quasi-perpendicular and the quasi-parallel bow shock currents is displayed in Fig. 6 in the IMF-aligned coordinate systems. The current directions are represented by arrows of normalized length and the colour code indicates the direction of currents with respect to the $x$ axis with red arrows pointing towards Earth and green arrows pointing towards sun. From theory, within this coordinate system the currents are expected to possess positive $y$ components, independently from their locations. As visible in the $y$-$z$-projections the flow directions are prescribed by the IMF orientation very clearly as they are collectively pointing along the positive $y$ axis in the IMF-aligned coordinate system. The histograms of the $J_{yz}$ direction in the bottom panel of Fig. 6 illustrates this in a quantitative manner. It shows a very clear peak around $0°$ in the quasi-perpendicular case. The peak is also visible in the quasi-parallel case, but it is much broader and currents with negative y components are observed frequently.

Having a closer look at the color code and the x-y-projection reveals that the currents are following the draped shape of the bow shock pointing towards sun and Earth at the flanks. Again, more deviations are visible during quasi-parallel bow shock crossings (Fig. 6, right panels). Hamrin et al. (2017) investigated $J_x$ in a similar way. As their events are located near the bow shock nose because of the MMS orbit, they introduced the approximation $J_x \approx J_n$, where $J_n$ depicts the portion of the current that flows normal to the shock surface. Their events lie within a range of about 7 Earth radii distance from the Earth-sun-line which approximately corresponds to a distance of about $0.5\,\Delta_{\mathrm{BS}}$ in our coordinate system. Near the bow shock nose they find, that $J_x$ points towards sun at y < 0 (GSM) and toward Earth at y > 0 (GSM) for northward IMF. In case of southward IMF the current directions are reversed. The distribution of the colors in fig. 6 shows, that the results of the orientation of the $J_x$ component from the study conducted by Hamrin et al. (2017) and from our study are qualitatively identical and are also valid for larger distances from the bow shock nose.

In order to investigate the current normal component in our study, we calculate $J_n$ via the local normal direction of the model bow shock surface for all quasi-perpendicular events. We analyze the orientation of the normal component in dependence of the $y$ coordinate within the IMF-aligned coordinate system). Table 1 gives the occurrence rate of outward and inward orientation of $J_n$ at a region close to the bow shock nose, approximately corresponding to the location of events within the study by Hamrin et al. (2017) as well as a region further away from the nose at the flanks of the bow shock. Based on our results, we can not identify a general dependence of the $J_n$ orientation from location of the events. Overall, the currents are slightly more often pointing outwards.

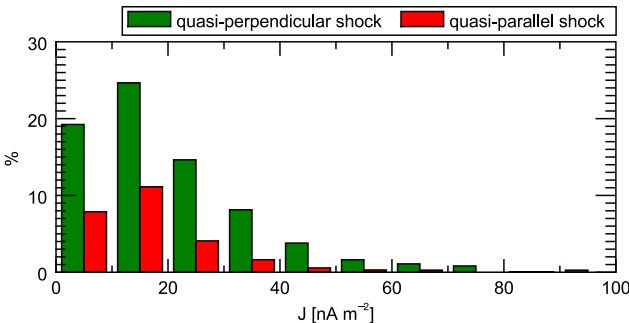

**Figure 7.** Occurrence rate of bow shock current magnitudes with a $10\,\mathrm{nA\,m^{-2}}$ binning interval.

## 4.2 Current magnitudes

Figure 7 shows the occurrence distribution of investigated current magnitudes. The majority (about 80 %) does not exceed $30\,\mathrm{nA\,m^{-2}}$ and the average current magnitude of all events is $19.4\,\mathrm{nA\,m^{-2}}$. As the current magnitudes calculated by the Curlometer are influenced by the averaging in time and space (averaging window, spacecraft separation, average current density along event trajectory), it is more likely that current magnitudes tend to be underestimated than overestimated. A direct comparison of some events which were analyzed in the study by Tang et al. (2012) as well as in our study show that the current density magnitudes calculated with the Curlometer technique are by a factor between 2.7 and 4.5 smaller than those calculated by determination of the layer thickness and the jump in the magnetic field.

MHD simulations (e.g. Lopez et al., 2011) predict a relatively broad region around the bow shock nose where the current magnitudes are constantly high, while the magnitudes are decreasing at the high latitude bow shock. Investigating the spatial dependence of the event's current magnitudes results in a roughly homogeneous distribution in case of the quasi-parallel shock events which are all located at low latitudes (cf. Fig. 6). In case of quasi-perpendicular events current magnitudes near the bow shock nose are slightly larger than those at the flanks. The average magnitude of all quasi-perpendicular currents below $60°$ latitude is $22.3\,\mathrm{nA\,m^{-2}}$ while the average value of currents at higher latitudes is $16.1\,\mathrm{nA\,m^{-2}}$.

The IMF magnitude is another controller of the bow shock current magnitude. Based on Ampère's law and the Rankine-Hugoniot conditions one can expect a linear correlation between the current magnitude and the magnetic field strength of the IMF tangential component with respect to the shock surface

$$J \propto [\boldsymbol{B}_t] \propto \mathbf{IMF}_t, \tag{4}$$

where the brackets denote the jump of the magnetic field across the discontinuity.

Tang et al. (2012) found this linear relation between bow shock current and $\mathrm{IMF}_z$ component at quasi-perpendicular bow shock events during northward and southward orientated IMF. Our survey shows that the linear correlation also applies to arbitrary IMF orientation and shock geometry. Figure 8 displays the average current magnitudes that are calculated for $1\,\mathrm{nT}$ intervals of the IMF component tangential to the bow shock surface. For quais-perpendicular and quasi-parallel cases magnetic

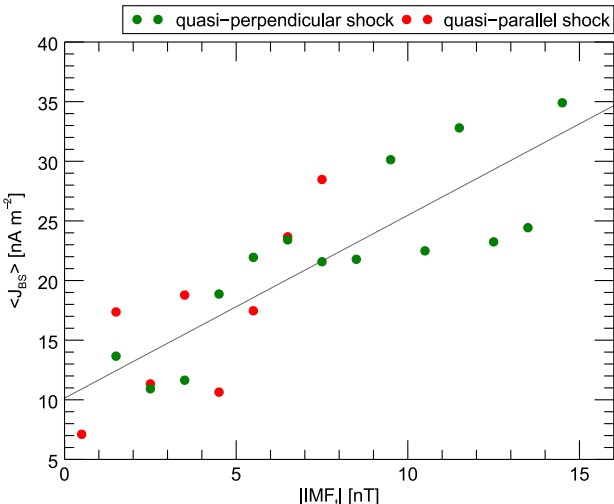

**Figure 8.** Current magnitude in dependence of the IMF tangential component. The $\text{IMF}_t$ component is calculated via the shock normal direction and is binned into intervals of 1 nT. The current magnitude $<J_{BS}>$ is calculated by averaging over all quasi-perpendicular and all quasi-parallel current events that are ascribed to the corresponding interval.

field values up to 15 nT and 8 nT are observed, respectively. Within the range from 0 to 8 nT there are no distinct qualitative differences between quasi-perpendicular and quasi-parallel situations visible.

In the limit of high Mach number one can derive

$$\boldsymbol{J} = 3\,\mathbf{IMF}_t/(\mu_0 L), \tag{5}$$

where $L$ is the bow shock thickness. The correlation coefficient of the linear fit within fig. 8 is 0.84. The slope of the fit provides an estimate of the average bow shock thickness of about 1600 km. As mentioned above, it is likely that the magnitudes tend to be underestimated by the Curlometer technique. The value of 1600 km therefore represents an upper estimate of the shock thickness. Bale et al. (2003) performed an extensive study of the bow shock thickness which gives a typical scale of a few hundreds of kilometers.

## 5 Conclusions

The usage of the Curlometer technique allows us a direct investigation of 369 current events recorded within the magnetic field data obtained by Cluster during 111 bow shock crossings in 2002, 2003 and 2004. In 274 cases the bow shock represented a quasi-perpendicular shock ($\Phi > 45°$). In 95 events a quasi-parallel ($\Phi < 45°$) shock was observed. It lies adjacent to the foreshock region where solar wind particles are reflected back into the solar wind along the IMF direction and cause upstream and downstream perturbations of the magnetic field configuration.

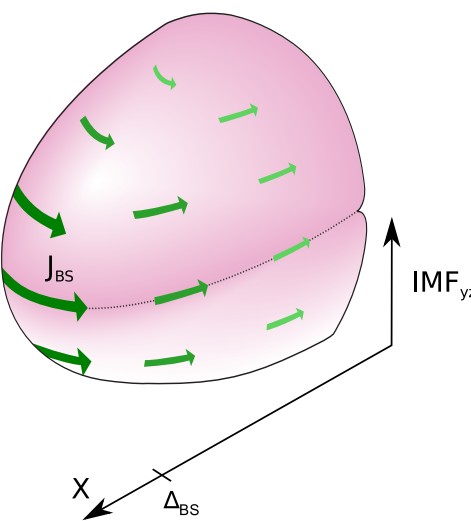

**Figure 9.** 3-D scheme of day-side bow sock shock current magnitude and orientation with respect to IMF *y-z*-component (GSM) in case of a quasi-perpendicular bow shock. Greater and darker arrows represent higher values of current magnitude.

At quasi-perpendicular shocks the bow shock currents are very clearly described by the IMF direction fulfilling the equation $\nabla \times \boldsymbol{B} = \mu_0 \boldsymbol{J}$. The angle distribution between the bow shock current and $\text{IMF}_\text{t}$ shows a sharp peak at $90°$. When displayed in an $\text{IMF}_\text{yz}$ aligned coordinate system the current directions arrows arrange themselves parallel to each other in the *y-z* plane. As the currents flow parallel to the shock surface, the draped shape of the bow shock becomes visible within the current direction

in the *x-y* plane. Current magnitudes are larger near the bow shock nose than at the flanks. Figure 9 is a schematic summary of our results for the currents observed at the day-side quasi-perpendicular bow shock.

The angle distribution between currents and the normal of the reference bow shock peaks at $90°$ as well but it is broadened to some extend as the reference bow shock naturally deviates from the true bow shock geometry. The magnitudes of currents at the quasi-perpendicular shock generally increase with increasing tangential component of the IMF. Typical values of the

averaged current magnitudes obtained by the Curlometer technique are in the range of 5 to $40\,\text{nA}\,\text{m}^{-2}$ with an average of about $20\,\text{nA}\,\text{m}^{-2}$. Those results are in the same order of magnitude but smaller than the ones determined in former studies by calculating the current density via the jump in the magnetic field and a derived current sheet thickness.

The quasi-parallel shocks that we have found are all located at relatively low latitudes. In this region, we were not able to observe a spatial dependence of the current magnitude. Additionally, the IMF tangential component posses lower values only

up to $8\,\text{nT}$. In this range we find that the dependence on the IMF magnitude seems to be qualitatively equal to that observed for quasi-perpendicular situations. In particular the current magnitudes of the quasi-perpendicular and the quasi-parallel bow shock are of similar size for a given IMF which is an interesting finding as the ideally (in reality never realized) parallel-shock would be accompanied by no jump in the magnetic field and therefore no current at all (Narita, 2006).

The direction of currents of the quasi-parallel bow shock are less described by the IMF orientation compared to the quasi-

perpendicular shock. Overall, the main characteristics are maintained but much more and larger deviations are visible. In

addition, the currents no longer lie perpendicular to the normal direction of the reference bow shock which indicates that the simplified model bow shock geometry does not hold at the quasi-parallel bow shock.

**Data availability**

All Cluster data used in this study can be retrieved from the Cluster Active Archive.

## 5 Competing interest

The authors declare that they have no conflict of interest.

*Acknowledgements.* This work was financially supported by the Deutsches Zentrum für Luft- und Raumfahrt under contract 50OC1402. The author thanks the Cluster FGM and CIS Teams and the Cluster Science Archive for processing and providing the Cluster data.

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
