# Peer review of "Statistical survey of day-side magnetospheric current flow using Cluster observations: Bow shock"

_Annales Geophysicae, 2018_

## Referee Comment (RC1) · Anonymous Referee #1 · 19 Mar 2018

I thank the Editor for choosing me to review this manuscript titled "Statistical survey of day-side magnetospheric current flow using Cluster Observations: Bow Shock". In this manuscript the authors estimated bowshock currents using curlometer method and compared with theoretical understanding of bowshock currents. To my knowledge this seems to be one of the early statistical studies which tried to verify our understanding of Bowshock current. Hamrin et al. (2017, https://doi.org/10.1002/2017JA024826) presented a statistical study on bowshock current closure using MMS data which is limited to low latitudes. However, this study verifies our theoretical understanding of bowshock currents from Cluster data which has highlatitude coverage aswell. I suggest the authors also comment on their view on bowshock current closure based on their dataset.

[Figure]

This will improve the manuscript's scientific quality. The authors in current manuscript presented the topic in a simple and logical manner. However, I have one general comment – Figure captions do not describe the figure properly. This makes the reader to expect more information in the text but I find the authors' description of figures in the manuscript text is brief. Below are some comments and suggestions which I think will improve the quality of this manuscript. 1) I suggest the authors add a figure on location of Cluster spacecraft during the events presented in this manuscript. Such a figure can be one similar to Figure 4 (top panel) with representative location of Cluster tetrahedra. It is important for the reader to understand the spatial coverage of results presented in this study. 2) Page 4, lines 1&2: The authors mention rotating GSE coordinate system to align with IMF yz components. I suggest explaining why the authors choose to do this and what is the advantage of such a rotation in the text. Also, in page 5, line 2, authors mentioned IMF-aligned coordinate system first time in this study. I guess they are referring to the coordinate rotation mentioned in page 4 but I suggest defining IMF-aligned system in page 4. 3) Description of Figure 6 in section 4.2 is not clear. The authors seem to say that a linear relation between bow shock current and IMF z-component is seen in Fig 6 which is in line with Tang et al. (2012). Looking at figure 6, I do not agree with this conclusion. I suggest the authors describe the text for figure 6 clearer. 4) Figure 7 presents a schematic of bowshock currents but did not describe the distance ranges that this schematic is valid. I suggest the authors describe how Cluster results presented in this paper support this schematic. Adding a figure of Cluster position as suggested in point 1 above would help understand Figure 7. 5) In description of Figure 5 (Page 5), the authors compared current magnitudes obtained in this study with those presented in Lopez et al. (2011) but they did not comment on current directions or the current closure. But the schematic in Figure 7 seems to suggest the bow-shock currents do close on themselves, is that right? If yes, explain how the results presented support your conclusion. Lopez et al. (2011) suggested that in MHD simulations, bowshock currents closed with magnetospheric currents. A recent study by Hamrin et al. (2017, https://doi.org/10.1002/2017JA024826) presented statis-

tical study on bow-shock current closure using MMS data but this study limited by the lack of high latitude bowshock crossings. I would suggest that the authors present their view of bowshock current closure based on their dataset as this Cluster dataset has wider latitudinal coverage than MMS study by Hamrin et al. (2017). Again, adding a figure with Cluster locations for events used in this study will help clarify this manuscript better. 6) I suggest the authors discuss Hamrin et al. (2017) in their introduction and where possible compare results presented in this study with those of Hamrin et al. (2017).

Some minor language issues: Page3, lines 1-2: This sentence seems to be a bit complicated. Suggest rewriting to make it simpler. Page 3, line 11: add "to" between approach and solving. Page 4, line 13 and elsewhere throughout: suggest using "deviates" or something similar instead of "diverts". Page 5, line 4: The current's –> The current Page 5, line 4 and elsewhere throughout: Use some other word like "described" instead of "prescribed"

Please also note the supplement to this comment:
https://www.ann-geophys-discuss.net/angeo-2018-9/angeo-2018-9-RC1-supplement.zip

---

## Referee Comment (RC2) · Anonymous Referee #2 · 13 Apr 2018

This manuscript presents observations of the Earth's bow shock with Cluster and the calculation of the shock electric current on more than 300 events. Statistics of the currents with their direction and strength has been performed and in particular the distinction between quasi-parallel and quasi-perpendicular bow shocks.

The manuscript is well written, the data are well presented and the conclusions are clear.

One main comment is how accurate is the current calculated for quasi-parallel shocks. Quasi-parallel shocks are more variable and "noisy" than quasi-perpendicular shocks which may make their current more difficult to measure. An example of crossing for

quasi-perp and quasi-para and their respective computation of current could be shown in a figure to introduce these differences.

Detailed comments:

p2 l30: it would be good to specify if the accuracies are on the current calculation.

p2 l38: 30s seems quite long, did you try if lower numbers changed the results?

p2 l50: database instead of data basis

p2 l75: what is the reason to rotate Xgse to have the IMFyz pointing in +Z direction? Just to have a coordinate system where the IMF is all in the same direction? Any reason why choosing point in the +Z direction?

p3 l4: I would add a comma after IMF

p3 l33: Is it Jz or Jyz

p3 l48: the sentence "A weakening .." is not clear

p3 l51: is "allocate" the right word? "are located" may be better.

p3 l60: northward and southward in one word.

p4 l5: it may be worst to investigate if a fit could be obtained from these points and how does it compare to Tang et al. and theory/simulations.

p5 l25: any idea why quasi para shocks are observed only at low latitude? In principle their observation should be similar to quasi-perp.

p5 l35: could some of the quasi-para shock be quasi-perp shock due to inaccuracy of IMF measurements or quick changes of IMF?

Figure 3 and 5: instead of adding the quasi-perp and quasi-para on top of each other, I would put them side by side in the same bin (to see more clearly the distribution for quasi-para shocks).

---

## Author Comment (AC1) · 16 Apr 2018

We want to thank the referee for this helpful and constructive criticism.

General comments:

Figure captions: Thanks for pointing out that several figure descriptions should be optimized for a better understanding of the pictures themselves as well as the results of our study. We will improve the figure captions and add more descriptions in the text.

Language issues: We will correct them in the revision.

Special comments:

[Figure]

1) Locations of events: We will add a picture presenting the locations of the investigated Cluster bow shock crossings in GSM-coordinates.

2) IMF-aligned Coordinate-System, rotation around x-axis: Thanks for pointing out that the description of the motivation for this transformation as well as the formulations in the text should be improved. The current directions at the bow shock are directly controlled by the IMF orientation via Amperè's law. As we do not confine our study to mainly north-south orientated IMF, the presentation of the resulting current directions in a GSE or GSM-system like in Fig. 4 but without the rotation around the x-axis leads to a quite chaotic looking distribution of the current arrows making it impossible to extract any useful information from it as indeed the required information of the IMF would not be included in such a picture. In contrast, by using the rotated system, aligned to the IMF-x-z-componten for each event, the collective orientation of the currents becomes visible as presented in Fig. 4. Also the differences between the quasi-parallel and the quasi-perpendicular bow shock currents are plain to see after the transformation. In our revision we will correct the existing inconsistency of the formulations regarding the use of our reference system and add additional words about the motivation and advantage of using it.

3) Based on Amperè's law and the Rankine-Hugoniot conditions one can expect a linear correlation between the current magnitude and the mangetic field strength of the IMF tangential component with respect to the shock surface: $J \sim [B\_t] \sim B\_{IMF,t}$. To make this clearer in figure 6 we will add a linear fit to the data points (see attached figure 1). The correlation coefficient is 0.84.

In the limit of high Mach number one can derive $J = 3 B\_{IMF,t}/(\mu 0 L)$ where L is the bow shock thickness. The slope of the linear fit provides an estimate of the average bow shock thickness of about 1600 km. As the current magnitudes are influenced by the averaging in time and space (averaging window, spacecraft separation, average current density along event trajectory), it is more likely that the current magnitudes tend to be underestimated than overestimated. A direct comparison of some events wich were
analyzed in the study by Tang et al. 2012 as well as in our study show that the current density magnitudes calculated with the curlometer technique are by a factor between 2.7 and 4.5 smaller than those calculated by determination of the layer thickness and the jump in the magnetic field. The value of 1600 km therefore represents an upper estimate of the shock thickness. Bale at al. 2003 performed an extensive study of the bow shock thickness which gives a typical scale of a few hundreds of kilometers.

4) We will add a more specific caption to the scheme, also mentioning the spatial scales.

5) Fig. 5 only gives information about the magnitudes. The current directions are discussed in the sections above accompanied by the figures 3 and 4.

Current closure: Thank you very much for your hint at the work of Hamrin et al., 2017, and their approach for investigating the current closure via the current component normal to the bow shock. That is a very interesting idea, and we have now performed a similar investigation of our bow shock currents. To enable a good comparison between the results from Hamrin et al. (their event selection focused on "clear and simple" events) and our events we restrict our analysis to the observed quasi-perpendicular events. Because of the MMS-orbit, the events within the Hamrin et al. study are mainly located within the range of about -7 RE < y < 7 RE (GSM). Transferred to the system of reference we use in our study this corresponds approximately to -0.5 D_BS < y < 0.5 D_BS, where D_BS is the bow shock standoff distance (compare fig. 4).

As the events from the Hamrin et al. study are located near the bow shock nose they introduce the approximation Jn = Jx. They find that the Jn (=Jx) components point outwards at y < 0 and inwards at y > 0 for northward IMF. This is consistent to the current direction parallel to the bow shock in the picture of possible current closure. For southward IMF all current directions are reversed.

Our picture 4 shows the Jx component of the Cluster events within the top panel (x-y-plane). Additionally, the Jx direction is presented by the color code used in the top

and the middle (y-z-plane) panel. Green colors depict Jx directions that are pointing outwards and red colors depict Jx directions that are pointing inwards. At y < 0 the geen color is dominating, while red dominates at y > 0. As the events are presented in our reference system (GSM rotated around x-axis), the picture includes northward, southward, and intermediate IMF orientations all together.

The distribution of the colors in fig. 4 shows that the results for the orientation of the Jx component from the Hamrin study and from our study are qualitatively identical. A significant difference is the spacial distribution of the events. The MMS events provide a very good coverage near the bow shock nose, while the Cluster events are distributed over wider distances from the bow shock nose. In our study, we interpreted the spatial distribution of the Jx component as a resemblence of the large scale bow shock curvature. In fig. 4, x-y-plane, one can see that the currents follow the shape of the model bow shock quite well.

The approximation Jn = Jx could include a significant error when applied to the Cluster events from our study because of the distance from the shock nose. In our additional investigation we therefore calculate Jn via the local bow shock normal: Jn = J * n We analyzed the orientation of Jn in dependence of the y-coordinate (within the reference system) for our quasi-perpendicular shock events. The table shown it attached figure 2 gives the occurance rate of outward and inward orientation for -0.5 D_BS < y < 0 and 0 < y < 0.5 D_BS (which is about the coverage of the MMS events from the Hamrin et al. study) as well as for y < -0.5 D_BS and 0.5 D_BS < y (additional locations because of the Cluster orbit)

Based on these numbers we can not identify a general majority of outwart pointing Jn at y < 0 and inward pointing Jn at y > 0 which would be expected from the picture of Jn resembling the current closure via the magnetosheath.

6) Results from Harmin et al., 2017: We will include references to as well as comparisons with their results in our revision.

[Figure]

[Figure]

**Fig. 1.**

Occurance rate of outward and inward pointing directions of the current normal component at the quasi-perpendicular shock depending on the y-coordiniate (reference system) in units of the bow shock standoff distance.

|                     | $y < -0.5$ | $-0.5 < y < 0$ | $0 < y < 0.5$ | $0.5 < y$ |
|---------------------|-----------|---------------|--------------|-----------|
| Jn pointing outwards | 38 (45%)  | 36 (61%)      | 37 (57%)     | 47 (67%)  |
| Jn pointing inwards: | 46 (55%)  | 29 (39%)      | 28 (43%)     | 27 (36%)  |

**Fig. 2.**

---

## Author Comment (AC2) · 25 Apr 2018

Response to Referee #2

We thank the referee for his constructive and helpful comments.

1) Comment on current calculation for quasi-parallel shocks:

We will add an example event to demonstrate the identification and calculation of the currents after the application of the curlometer tool. In our study the currents are identified directly by inspection of the curlometer results. The categorization into quasi-perpendicular and quasi-parallel shocks follows afterwards. Currents are identified as

bow shock currents when the following cirteria are fulfilled: 1) clear current peak within the curlometer results visible. 2) current event coincides with particle data signatures that are consistent with a bow shock crossing.

There are cases where the current or the particle data (or both) show very fluctuating behavior making it difficult to identify current structures at a bow shock crossings. In cases, where the identification becomes presumably unreliable, the events were omitted and not included in our study.

The categorization into quasi-perpendicular and quasi-parallel geometries in the following step revealed a significant imbalance. Almost three times more quasi-perpendicular events were identified. It is very likely that this imbalance is caused especially by the fact, that the shocks are more "noisy", fluctuating, and less "clear" in the quasi-parallel case and were therefore more often omitted.

2) Accuracies of the current determination:

The accuracies are mentioned at p2 l25.

3) 30 sec winow, results with other windows:

We tried different sized averaging windows. The direction of the currents prove to be quite insensitive to a window size between about 20 and 40 sec. When the window is chosen too long, for example above 60 sec at the bow shock, the directions alter significantly. For small windows below 10 sec the variations within the current direction determination rise as well because of the influence of small scale fluctuations.

The current magnitudes are more sensitive to the averaging window. The damping of high frequency flucutations and the influence of the associated spatial averaging directly leads to smaller current peak magnitudes calculated by the curlometer. This effect is less intense when the average magnitudes per event are calculated. In this study we analyze the average magnitudes instead of the peak magnitudes.

The window size also influences the current identification. For very small windows,

surrounding high frequency fluctuations lead to additional very sharp current peaks which reach similar magnitudes as the crossing itself, making it harder to identify the correct current structure associated with the crossing. Recurring features resembling a series of non-similar delta functions or single delta functions are filter out by the application of the averaging window. Step functions (resembling a simple "clear" bow shock transition) are not filtered out but are maintained less sharp which mainly affects the current density's peak magnitude. Recurring features that can be interpreted as a series of similar delta functions (for example multiple encounters with a bow shock of stable configuration within a short time) are merged by the windowing process and maintained too. The described effects of the windowing becomes clear for example in the attached figure 1 which is taken from our previous publication. It shows a comparison of 0 sec and 100 sec averaging at multiple magnetopause crossing.

With too small averaging windows, the reliability of the curlometer results becomes small as well, as the basic assumptions of the curlometer technique are violated to a larger extent. Overall, the 30 sec window proved to be a practical and reliable choice for the bow shock current investigation.

4) IMF-aligned Coordinate-System, rotation around x-axis:

Thanks to both referees for pointing out that the description of the motivation for this transformation should be improved. The current directions at the bow shock are directly controlled by the IMF orientation via Amperè's law. As we do not confine our study to mainly north-south orientated IMF, the presentation of the resulting current directions in a GSE-system like in Fig. 4 but without the rotation around the x-axis leads to a quite chaotic looking distribution of the current arrows making it impossible to extract any useful information from it as indeed the required information of the IMF would not be included in such a picture. In contrast, by using the rotated system, aligned to the IMF x-z-component for each event, the collective orientation of the currents becomes visible as presented in Fig. 4. Also, the differences between the quasi-parallel and the quasi-perpendicular bow shock currents are plain to see after the transformation. In

our revision we will add additional words about the motivation and advantage of using it. In principle, the choice of aligning the IMF to the positive z-direction is arbitrary. But as it is quite common in other publications to distinguish especially between northward and southward IMF and to use pure northward and southward IMF configurations as basic situations for magnetospheric simulations, it seemed natural to us, to chose the positive z-direction.

5) Data fit, p4 l5:

Probably you are referring to fig 6, which is located on page 7 in the preview version we have access to. We will add a linear fit to the data points (see attached figure 2). The correlation coefficient is 0.84.

In the limit of high Mach number one can derive J = 3 B_IMF,t/($\mu$0 L) where L is the bow shock thickness. The slope of the linear fit provides an estimate of the average bow shock thickness of about 1600 km. As the current magnitudes are influenced by the averaging in time and space (averaging window, spacecraft separation, average current density along event trajectory), it is more likely that current magnitudes tend to be underestimated than overestimated. A direct comparison of some events which were analyzed in the study by Tang et al. 2012 as well as in our study show that the current density magnitudes calculated with the curlometer technique are by a factor between 2.7 and 4.5 smaller than those calculated by determination of the layer thickness and the jump in the magnetic field. The value of 1600 km therefore represents an upper estimate of the shock thickness. Bale at al. 2003 performed an extensive study of the bow shock thickness which gives a typical scale of a few hundreds of kilometers.

6) No quasi-parallel shocks at high latitudes:

We have not investigated this effect yet. From basic theory it is expected, that the os-cillations and variations which are triggered at the quasi-parallel shock develop within the foreshock region while they are convected towards the shock. At higher latitudes, there is more space/time available for this development before the bow shock is en-
countered. Because of that one can expect stronger disturbances and variations at the quasi-parallel bow shock at higher latitudes than at lower ones. As mentioned above, we only use events within our study, where a reliable identification of the shock current was possible. It might well be that high latitude quasi-parallel events were unintentionally omitted due to this restriction. Still, this is only speculation as we have not studied this in detail yet.

7) Accuracy of distinction between quasi-perpendicular and quasi-parallel cases:

It is possible, that some events are not categorized correctly because of uncertainties within the determination of the shock normal and the IMF direction, mainly when the angle is near 45°. This leads to some quantitative uncertainties within the results. As the differences between the quasi-perpendicular and the quasi-parallel events are still very obvious, these uncertainties to not significantly alter the qualitative results.

8) Rearragement of bars within figures 3 and 5:

Thank you for this suggestion. Putting them side by side makes the distribution for the quasi-parallel events much clearer.

9) Language issues:

Thanks for pointing them out. We will correct them in the revision.
* * *
2004 Mar 18

**Fig. 1.**

[Figure]

Fig. 2.